# Isolation and characterization of a covalent Ce$^{IV}$-Aryl complex with an anomalous $^{13}$C chemical shift

Grace B. Panetti [1], Dumitru-Claudiu Sergentu [2], Michael R. Gau[1], Patrick J. Carroll[1], Jochen Autschbach [2✉], Patrick J. Walsh [1✉] & Eric J. Schelter [1✉]

The synthesis of bona fide organometallic Ce$^{IV}$ complexes is a formidable challenge given the typically oxidizing properties of the Ce$^{IV}$ cation and reducing tendencies of carbanions. Herein, we report a pair of compounds comprising a Ce$^{IV} - C_{aryl}$ bond [Li(THF)$_4$][Ce$^{IV}$($\kappa^2$-*ortho*-oxa)(MBP)$_2$] (**3-THF**) and [Li(DME)$_3$][Ce$^{IV}$($\kappa^2$-*ortho*-oxa)(MBP)$_2$] (**3-DME**), *ortho*-oxa = dihydro-dimethyl-2-[4-(trifluoromethyl)phenyl]-oxazolide, MBP$^{2-}$ = 2,2′-methylenebis(6-*tert*-butyl-4-methylphenolate), which exhibit Ce$^{IV} - C_{aryl}$ bond lengths of 2.571(7) – 2.5806(19) Å and strongly-deshielded, Ce$^{IV} - C_{ipso}$ $^{13}$C{$^1$H} NMR resonances at 255.6 ppm. Computational analyses reveal the Ce contribution to the Ce$^{IV} - C_{aryl}$ bond of **3-THF** is ~12%, indicating appreciable metal-ligand covalency. Computations also reproduce the characteristic $^{13}$C{$^1$H} resonance, and show a strong influence from spin-orbit coupling (SOC) effects on the chemical shift. The results demonstrate that SOC-driven deshielding is present for Ce$^{IV} - C_{ipso}$ $^{13}$C{$^1$H} resonances and not just for diamagnetic actinide compounds.

[1] Roy and Diana Vagelos Laboratories Department of Chemistry, University of Pennsylvania, Philadelphia, PA, USA. [2] Department of Chemistry, University at Buffalo, State University of New York, Buffalo, NY, USA. ✉email: jochena@buffalo.edu; pwalsh@sas.upenn.edu; schelter@sas.upenn.edu

The bonding between lanthanides and ligands has been described as purely ionic[1,2]. However, modern spectroscopic and computational techniques have challenged this simple assumption[3–6]. The covalency in M − X interactions can result in anomalous and diagnostic nuclear magnetic resonance shifts, X = $^{13}C$, $^{15}N$, $^{19}F$, $^{77}Se$, $^{125}Te$, resulting from participation of f-element orbital angular momentum[7–13]. Another important aspect of detailed f-element electronic structure is multiconfigurational character, as described in the model example of cerocene, Ce(COT)$_2$, COT = cyclooctatetraene ligand[14]. The case for multiconfigurational character in cerocene has been made through X-ray absorption, SQUID magnetometry, and multireference computational studies and tied strongly to the characteristics of the cerium-carbon bonding[15]. Despite the interest surrounding f-element covalency and multiconfigurational effects, there are few examples of organometallic Ce$^{IV}$ complexes[16]. The current literature is limited to metallocene Ce$^{IV}$ complexes or α-heteroatom stabilized Ce$^{IV}$ − C σ-bonds. Reported examples of Ce$^{IV}$ − metallocene bonding include Ce$^{IV}$ complexes of cyclopentadienide, cyclooctatetraene dianion, and bispentalene dianion ligands[17–21]. Complexes containing a Ce$^{IV}$ − C σ-bond, however, are limited to either an N-heterocyclic carbene (NHC) complexes, e.g., Ce[L$_4$] (Fig. 1a), or a bis(iminophosphorano)methandiide complex, e.g., [Ce(BIPM$^{TMS}$)(ODipp)$_2$] (Fig. 1a)[22,23]. The latter examples are expected to have electronic structures that deviate significantly from typical organometallic alkyl, aryl, or alkynyl ligands due to the heteroatom α-substitution[22–25]. The scarcity of Ce$^{IV}$ − C containing-complexes likely arises from the unstable combination of strongly reducing carbanions and the oxidizing Ce$^{IV}$ cation[16]. As a result, the formation of reactive, carbon-centered radicals and Ce$^{III}$ species is observed. Our team has a long-standing interest in the isolation of redox stable Ce$^{IV}$ species to elucidate the relationship between ligand field and the Ce$^{III}$/Ce$^{IV}$ couple[26–28].

Herein, we expand our studies for the isolation of a pair of Ce$^{IV}$ − C$_{aryl}$ compounds. These compounds display unusually high $^{13}C$ NMR shifts compared to other diamagnetic M$^{IV}$ − C$_{aryl}$ compounds. Relativistic density functional calculations verify that the high NMR shifts are due to large SOC effects supported by the increased covalency of the Ce$^{IV}$ − C$_{aryl}$ bond.

## Results
### Synthesis and structures of Ce$^{IV}$ − C$_{aryl}$. Considering strategies to stabilize a Ce$^{IV}$ − C$_{aryl}$ bond, we hypothesized that tethering

the aryl group to the Ce center would kinetically inhibit homolysis of the Ce − C bond. In addition, we sought a sterically-protected Ce center to prevent reactivity at the *ipso*-carbon. Lastly, we chose a supporting ligand that would stabilize the Ce$^{IV}$ oxidation state to prevent charge transfer and subsequent Ce − C bond homolysis. With these considerations in mind, we aimed to prepare a Ce$^{IV}$ − C$_{aryl}$ bond from the Ce$^{IV}$ bis(methylene bisphenolate) complex Ce(THF)$_2$(MBP)$_2$, that was previously synthesized by members of the Schelter laboratory (**1**, Fig. 2)[29]. Aryloxide ligands have been previously shown to both stabilize the Ce$^{IV}$ oxidation state and high valent organometallic species of other metal species[28–34]. Addition of a yellow solution of *ortho*-lithiated oxazoline **2** (Li-*ortho*-oxa) to a purple benzene solution of Ce(THF)$_2$(MBP)$_2$ (**1**) at room temperature resulted in an immediate color change of the solution to dark red. The $^1H$ NMR spectrum of the reaction mixture revealed loss of the pseudo C$_{2v}$ symmetry of **1** and formation of a C$_1$ symmetric product. Likewise, there was also a shift in both the $^7Li$ and $^{19}F$ NMR resonances of **2**, and the two methylene protons and methyl groups of the oxazoline were no longer degenerate in the $^1H$ NMR spectrum. All $^1H$, $^7Li$, and $^{19}F$ NMR resonances were well within the range of diamagnetic signals, leading to the assignment of the product of the reaction as [Li(THF)$_4$][Ce(κ$^2$-*ortho*-oxa)(MBP)$_2$] (**3-THF**). Dark red X-ray quality crystals of **3-THF** were grown over 3 days from a cooled (−25 °C) mixture of **3-THF** in toluene and THF layered with pentane. The crystals were collected in 66% yield (Fig. 2). Alternatively, crystallizing from a cooled (−25 °C) solution of crude **3-THF** in DME layered with pentane resulted in dark-red X-ray quality crystals of **3-DME** over 3 days (Fig. 3). Crystals of **3-DME** were collected in a slightly higher 75% yield. The differences in NMR data between **3-THF** and **3-DME** are negligible compared to experimental error. While compound **3-DME** crystallizes with a single molecule in the asymmetric unit, compound **3-THF** crystallizes with two independent molecules in the asymmetric unit, with only minor differences between the structures. The Ce − O(phenoxide) bond distances of **3-THF** and **3-DME** (2.1636(13)–2.202(4) Å) compare well with the Ce − O (phenoxide) bond distances observed in the reported structure of **1** (2.113(2)–2.152(2) Å)[29]. The Ce − C bond distances of **3-THF** and **3-DME** are 2.571(7) – 2.5806(19) Å and are shorter than reported Ce$^{III}$ − C$_{aryl}$ bond lengths: 2.621(4) – 2.64 ± 0.02 Å[35,36]. The difference in ionic radii between 6-coordinate Ce$^{III}$ and Ce$^{IV}$ is 0.14 Å; however, the difference between **3-THF** and **3-DME** and previous Ce$^{III}$ − C$_{aryl}$ complexes is only 0.04 – 0.07 Å[37].

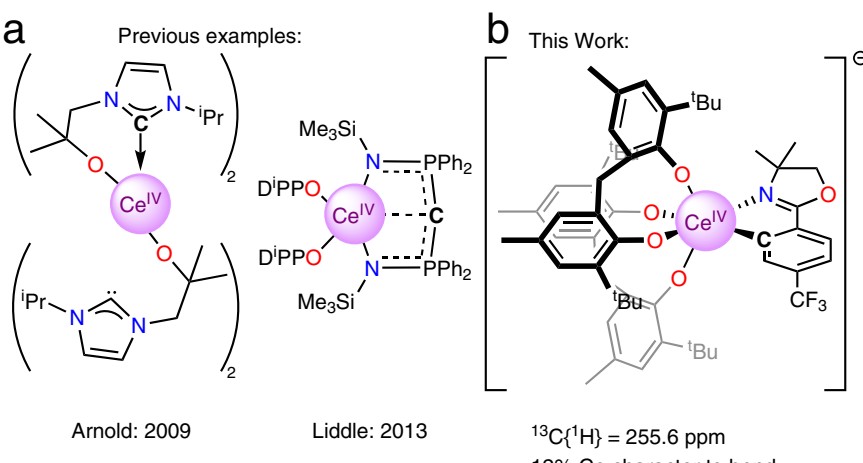

**Fig. 1 Examples of Ce$^{IV}$ − C σ bonds. a** Previous examples of complexes with formally Ce$^{IV}$ − C σ bonds, which are stabilized by either nitrogen[22] or phosphorus heteroatoms[23]. **b** This work detailing the synthesis and characterization of a Ce$^{IV}$ − C$_{aryl}$ bond, including computational analysis. Carbon atoms bound to cerium are indicated with a **C**.

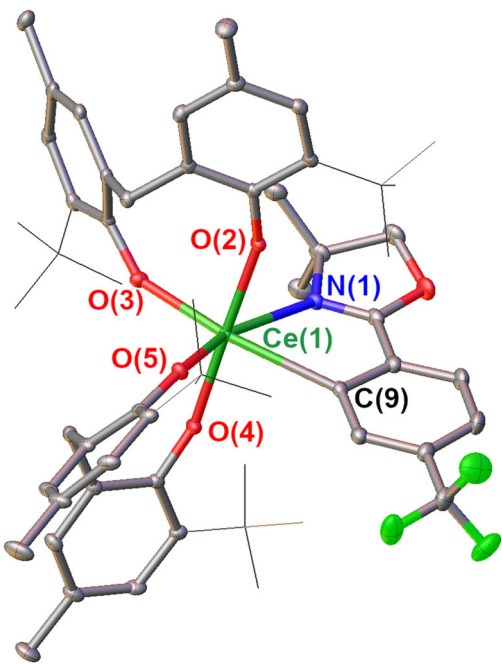

**Fig. 2 Syntheses of title $Ce^{IV} - C_{aryl}$ compounds 3-THF and 3-DME.** The complex **1** was treated with the isolable aryl lithium **2** to generate the title complexes **3-THF** or **3-DME**.

**Fig. 3 Crystal structure of 3-DME.** Thermal ellipsoid plot of the X-ray crystal structure of **3-DME** at the 30% probability level. For clarity, C–H hydrogens and the Li(DME)$_3^+$ cation were removed; In addition *tert*-butyl groups are displayed in wireframe. Selected bond lengths in Å: Ce(1)-C(9): 2.5806(19); Ce(1)-N(1): 2.6176(16); Ce(1)-O(2): 2.1750(12); Ce(1)-O(3): 2.2062(13); Ce(1)-O(4): 2.1640(12); Ce(1)-O(5): 2.1636(13).

We ascribe this difference to the steric demand by the MTB ligands vs. the pentamethylcyclopentadienyl ligands used in the prior work. Previously reported complexes containing $Ce^{IV} - C$ σ-bonds are 2.652(7) – 2.705(2) Å and 2.385(2) – 2.441(5) Å for the Ce − C NHC and Ce − C(bis(iminophosphorano)methan-diide) ligands, respectively[22,23,25]. With this data in hand, we assign this complex as a $Ce^{IV} - C_{aryl}$ complex.

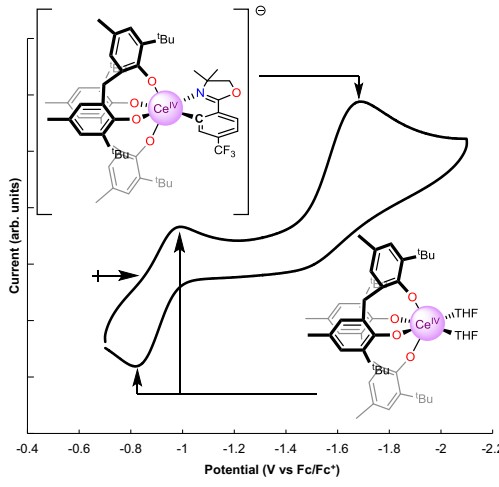

**Fig. 4 Cyclic voltammagram of 3-THF.** Solvent: THF; electrolyte 0.1 M [$^{n}Pr_4N$][$BAr^F_4$]; [analyte] = 0.001 M; OCP = −0.815 V vs. Fc/Fc⁺, noted by the right-facing arrow; $\nu$ = 100 mV s⁻¹. The trace shows the reduction of **3** at $E_{pc}$ = −1.67 V vs. Fc/Fc⁺ with a return wave at $E_{1/2}$ = −0.94 V vs. Fc/Fc⁺, which is the $Ce^{III}/Ce^{IV}$ couple of **1**[29]. The waves corresponding to **1** do not appear in the first scan (Supplementary Figs. 18–19).

**Electrochemical analysis.** To better understand how the *ortho*-oxa⁻ group impacts the stability of the $Ce^{IV}$ cation, electrochemistry was performed on **3-THF**. The $E_{pa}$ of **3-THF**, −1.67 V vs. Fc/Fc⁺, shifts by −0.72 V relative to the $E_{1/2}$ of **1** (−0.94 V vs. Fc/Fc⁺), indicating that the *ortho*-oxa⁻ moiety significantly stabilizes the $Ce^{IV}$ couple in THF. The reduction of **3-THF** is not reversible under the electrochemical conditions, although the event precedes a reversible oxidation at $E_{1/2}$ = −0.94 V vs. Fc/Fc⁺ and an irreversible oxidation at $E_{pa}$ = −0.43 V (Supplementary Figs. 18–19). We postulate that the reduction of the $Ce^{IV}$ center is followed by dissociation of the *ortho*-oxa⁻ fragment, producing **1** and **2**. Indeed, the return anodic scan comprises waves at $E_{1/2}$ = −0.94 V vs. Fc/Fc⁺ and $E_{pa}$ = −0.43 V respectively, consistent with the previous assignment for compound **1** and inferred for compound **2** (Fig. 4)[29].

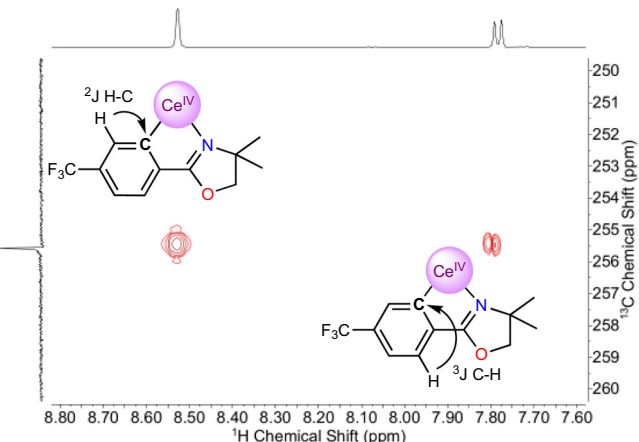

**Fig. 5 Identification of the unusual $^{13}$C resonance.** A portion of the HMBC spectrum showing the long-range C-H coupling to the $^{13}$C resonance at 255.6 ppm. An interpretation of the spectrum is inlaid, indicating that the signal at 255.6 ppm corresponds to the *ipso*-carbon.

### $^{13}$C NMR analysis.

While the $^1$H, $^7$Li, and $^{19}$F NMR of **3-THF** and **3-DME** showed minimal changes compared to the starting materials, the $^{13}$C{$^1$H} NMR of the *ipso*-carbon of both **3-THF** and **3-DME** showed a significant shift (difference between **3-THF** and **3-DME** is less than >0.05 ppm which is within error). Indeed, the *ipso*-$^{13}$C{$^1$H} resonance was located at 255.6 ppm, well outside of the typical range for aryl resonances (100–170 ppm) and shifted by ~50 ppm downfield relative to the $Li - C_{aryl}$ resonance for the starting material, **2** (Fig. 5). This $^{13}$C{$^1$H} shift is further downfield than observed for other characterized, diamagnetic $M^{IV} - C_{aryl}$ interactions; the highest being Th(2-C$_6$H$_4$CH$_2$NMe$_2$)$_4$, which exhibits a Th$^{IV} - C$ $^{13}$C{$^1$H} signal at 230.8 ppm[38,39]. The shift of the $^{13}$C{$^1$H} signal for the carbon atom bound to metal cations has been implicated as a reporter for the degree of covalency in f-element cation-carbon bonds[7–9]. In this light, **3-THF** and **3-DME** have an anomalously high covalency for a $M^{IV}$-aryl interaction. As with the bond distance metrics, there are few salient examples of $Ce^{IV} - C$ σ-bonds for comparison of the $^{13}$C{$^1$H} NMR shifts. The compounds isolated by P. Arnold and co-workers display a $^{13}$C{$^1$H} shift at ~213 ppm for the $Ce^{IV}$–NHC[22]. The compounds reported by Liddle contain $Ce^{IV} - C$ (bis(iminophosphorano) methandiide) $^{13}$C{$^1$H} shifts in the range of 324.6–343.5 ppm, depending on the secondary ligands bound to the $Ce^{IV}$ cation; [Ce (BIPM$^{TMS}$)(ODipp)$_2$] exhibits a $^{13}$C{$^1$H} shift of 324.6 ppm[23,25]. Notably, these compounds contain substantially different substituents attached to the Ce–$C$ carbon, diminishing the significance of their comparison.

### Computational bonding analysis.

To further understand the nature of the $Ce^{IV} - C_{aryl}$ interaction, we turned to computations to assess the electronic structure of the anionic, cerium-containing portion of **3-THF** (referred to as **3**). The geometry of **3** was optimized starting from the structure of **3-THF** determined by X-ray crystallography, using density functional theory (DFT) with the B3LYP functional, all-electron Slater-type basis sets for all atoms, and other standard settings as detailed in the SI. The agreement between experiment and theory was excellent, with only minor differences in the $Ce^{IV} -$ ligand bond lengths (≤0.02 Å). The MOs with the most Ce 4f character remain largely metal-centered and span the seven lowest unoccupied molecular orbitals (LUMO to LUMO + 6, Supplementary Figs. 13–19) of the complex, a common feature for $Ce^{IV}$ compounds as well as for cerium species with a debated $Ce^{IV}/Ce^{III}$ oxidation state[3,15]. HOMO to HOMO − 3 (Supplementary Figs. 26–29), for **3**

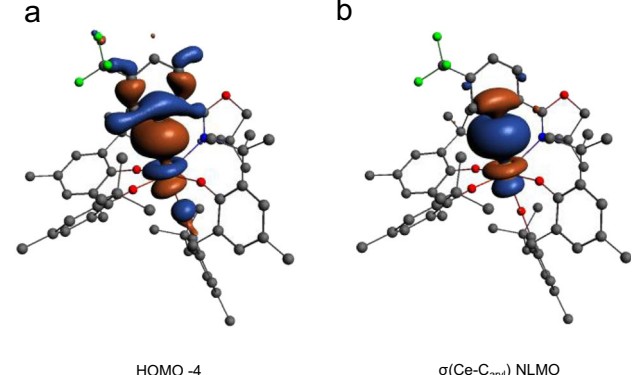

**Fig. 6 MO's of 3 depicting the Ce–C$_{aryl}$ bond. a** The DFT/B3LYP frontier Kohn–Sham molecular orbital of **3** (HOMO − 4). **b** The two-center two-electron bonding NLMO. Atomic orbital contributions of the NLMOs and other bond properties of all the Ce containing bonds are listed in Table 1.

**Table 1 NLMO compositions and bond orders for 3.**

| Bond/NLMO | %Ce (%s; %d; %f) | WBO[a] | MBO[b] | DI$^{QTAIM[c]}$ |
|---|---|---|---|---|
| σ(Ce-C) | 12 (5; 32; 62) | 0.41 | 0.46 | 0.50 |
| σ(Ce-N) | 4 (14; 49; 37) | 0.15 | 0.11 | 0.25 |
| σ(Ce-O)[d] | 3 (14; 50; 37) | 0.41 | 0.46 | 0.62 |
| 2xπ(Ce-O)[d] | 4 (1; 38; 61) | | | |

[a]Wiberg bond order in the natural atomic orbital (NAO) basis.
[b]Mayer–Mulliken bond order.
[c]Delocalization index based on QTAIM Bader analysis.
[d]The data are given as averages over the four Ce-O bonds.

are mostly delocalized phenoxide-centered orbitals, while HOMO − 4 (Fig. 6) corresponds to the highest occupied orbital showing significant ligand-metal ($Ce^{IV} - C_{aryl}$) hybridization.

The metal-ligand bonding in **3** is characterized in Fig. 6 and Table 1 in terms of natural localized molecular orbitals (NLMOs) and bond orders. There are two two-center two-electron σ bonds describing the donation bonding between the aryl carbon and oxazolinide nitrogen and Ce, and σ + 2π bonds describing the bonding between each of the O atoms and Ce (Supplementary Figs. 37–40). Among these, the $Ce^{IV} - C_{aryl}$ σ bond shown in Fig. 6 has the largest covalency, in terms of shared density, with 12% Ce contribution of which 32/62% involve 5d/4f. Previously reported $Ce^{IV} - C$ bonds contain 8–13% Ce contribution[25]. Ce tends to be less covalent than $U^{VI}$ ($U^{VI} - C$ bonds 22–29% U) but similar to Th$^{IV}$ (Th$^{IV} - C_{aryl}$ bonds 10–15%)[9,25,38,40]. In the remaining σ and π bonds with the N and O atoms, Ce contributes about 3–4% weight, suggesting that these bonds are mainly ionic. The bond ionicities are also reflected in the bond orders compiled in Table 1, all being significantly smaller than one (which would indicate a full single bond). In the sum of all interactions, however, Ce receives sizable electron donation from the surrounding ligands. For instance, the calculated Ce charge is +2.26 (Mulliken charge), +2.46 (Bader charge), and +2.44 (natural charge from a natural bond orbital (NBO) analysis) instead of the formal +4. The Ce natural electron configuration retrieved from the NBO analysis is 4f$^{0.76}$d$^{0.60}$, which deviates considerably from the formal 4f$^0$5d$^0$. The large Ce 4f electron count of **3** (0.76), associated mainly with the sizable Ce − C$_{aryl}$ bonding, is comparable to the calculated and experimentally-determined Ce 4f electron counts in CeO$_2$ and Ce(C$_8$H$_8$)$_2$[41–43]. We anticipate that this similarity has important implications

**Table 2 NLMO contributions to the $^{13}$C isotropic nuclear shielding ($\sigma_{iso}$) in 3[a].**

| NLMO[b] | $\sigma_{iso}^{SR}$, SR-ZORA | | | $\sigma_{iso}^{SO}$, SO-ZORA | | | $\Delta^{SO}$ [d] | | |
|---|---|---|---|---|---|---|---|---|---|
| | L[c] | NL[c] | L + NL | L | NL | L + NL | L | NL | L + NL |
| $\sigma$(Ce-C$_{aryl}$) | −103 | 2 | −102 | −156 | 2 | −154 | −52 | 0 | −52 |
| $\Sigma\sigma$(C$_{aryl}$-C$_{1,2}$) | −106 | −5 | −110 | −94 | −5 | −99 | 12 | −1 | 11 |
| 1s$_{core}$ (C$_{aryl}$) | 201 | 0 | 201 | 202 | 0 | 202 | 1 | 0 | 1 |
| $\Sigma_{other}$ | −26 | 11 | −15 | −25 | 11 | −14 | 1 | 0 | 1 |
| $\Sigma$(all of above) | −34 | 8 | −26 | −73 | 8 | −65 | −38 | −1 | −39[e] |

[a]DFT/PBEh-40 calculations. SR = scalar relativistic. SO = relativistic calculation including SOC. All shielding contributions are in ppm. The geometry orientation is such that the $^{13}$C corresponds to the origin of the cartesian axes and C–Ce bond aligns with the z-axis.
[b]The NLMOs are shown in Supplementary Fig. 41.
[c]L and NL indicate contributions from the Lewis and non-Lewis parts of the NLMO.
[d]Defined as $\sigma_{iso}^{SO} - \sigma_{iso}^{SR}$.
[e]SOC effects of +1/−39 ppm on the reference/probe shielding cause the +40 ppm SOC change in the chemical shift quoted in the text.

regarding the electronic structure of **3**, in the sense that it may potentially exhibit a multi-configurational ground-state wave-function with Ce$^{III/IV}$ character, similar to cerocene. However, further spectroscopic studies are needed, and are under way, to confirm this assignment for **3**.

**Computational chemical shift analysis**. Computed $^{13}$C NMR chemical shifts for the *ipso*-carbon, with various approaches, are compiled in Supplementary Table 2. The NMR shift was sensitive to the applied DFT approximations, a common observation in NMR shift calculations for compounds containing lanthanides and actinides. The best agreement with the experiment (256 ppm) for the *ipso*-$^{13}$C chemical shift in **3** was obtained with a PBE hybrid with 40% exact exchange, PBEh-40, which gave 259 ppm. The same functional previously provided excellent ligand chemical shifts in actinide complexes[40,44]. PBEh-40 produced a similar cerium electronic structure (Mulliken/natural charge of +2.43/+2.66, 4f$^{0.58}$5d$^{0.59}$ NBO natural electron configuration, Ce − C$_{aryl}$ WBO of 0.37) as B3LYP. Reasonable agreement with the experimental chemical shift was obtained also with the KT2 functional (265 ppm), which is known to perform well in NMR calculations[45]. The comparison between the *ipso*-$^{13}$C chemical shift calculated without and with SOC, with PBEh-40/KT2, reveals a 40/51 ppm downfield shift caused by SOC, which is largely triggered by the Ce 4f and 5d involvement in the Ce$^{IV}$ − C$_{aryl}$ σ bond.

In order to rationalize the anomalous $^{13}$C shift, we carried out an analysis of the DFT/PBEh-40 $^{13}$C isotropic shielding ($\sigma_{iso}$) in terms of NLMOs[46,47]. The NMR shielding data are gathered in Table 2 and the relevant NLMOs are shown in Supplementary Fig. 41. Note that these NLMOs are equivalent to those obtained with DFT/B3LYP. The analysis shows that the σ(Ce − C$_{aryl}$) covalent bond is the principal cause of the SOC-induced deshielding of the *ipso*-carbon. According to Table 2, the core orbital of the Ce-bound C$_{aryl}$ brings a diamagnetic contribution of roughly 200 ppm to $\sigma_{iso}$. This value is counterbalanced by two large, negative shielding contributions generated by the σ bonds involving the *ipso*-carbon, namely the σ(Ce − C$_{aryl}$) bond and the two σ(C$_{aryl}$ − C) bonds involving the two neighboring carbons of the aryl ring, denoted as C$_1$ and C$_2$. From Table 2 it is evident that the σ(Ce − C$_{aryl}$) contribution to $\sigma_{iso}$ is as important (without SOC) or even more important (with SOC) than the combined $\sigma_{iso}$ contributions generated by the σ(C$_{aryl}$ − C) NLMOs.

Indeed, with SOC in particular, the σ(Ce − C$_{aryl}$) NLMO yields a dominant negative contribution to $\sigma_{iso}$ of −154 ppm, of which −169 ppm is paramagnetic and 15 ppm is diamagnetic (not shown separately in Table 2), while the combined $\sigma_{iso}$ contribution from the σ(C$_{aryl}$ − C$_{1,2}$) NLMOs is −99 ppm. The overall SOC effects on the total isotropic shielding, $\sigma_{iso}$, add up to −39 ppm. The largest contribution to this value is generated by the σ

(Ce − C$_{aryl}$) NLMO, −52 ppm, and there are secondary contributions from the σ(C$_{aryl}$ − C$_{1,2}$) NLMOs, 11 ppm (Table 2, last column), indicating that these orbitals are somewhat delocalized onto the metal. Hence, the covalent bonding between the Ce center and C$_{aryl}$ is the main contributor of the $^{13}$C SOC deshielding and this aspect is strongly related to the sizable Ce 4f and 5d character of the σ(Ce − C$_{aryl}$) NLMO: 53% f and 41% d with PBEh-40, or 62% f and 32% d with B3LYP, with the 4f likely generating most of the SOC deshielding.

## Discussion

We have synthesized, characterized, and crystallized complexes featuring a Ce$^{IV}$ − C$_{aryl}$ bond. The synthesis of the title complexes **3-THF** and **3-DME** was realized from the reaction of the lithium-aryl salt **2** with **1**. Electrochemical analysis revealed that the aryl interaction resulted in a notable stabilization of the Ce$^{IV}$ oxidation state, shifting the $E_{pc}$ of Ce$^{IV}$ reduction by 720 mV relative to the THF adduct. A combination of $^{13}$C{$^1$H} NMR and DFT was used to investigate the covalency of the Ce$^{IV}$ − C$_{aryl}$ bonding. $^{13}$C{$^1$H} NMR analysis revealed that the *ipso*-carbon was shifted to 255.6 ppm, an indicator of metal-ligand covalency in the Ce$^{IV}$ − C$_{aryl}$ bond. This result was supported by NLMO analysis, which showed a 12% metal contribution to the Ce$^{IV}$ − C$_{aryl}$ bond. We expect that these results will further inform fundamental bonding in high valent f-elements and be effective in guiding the preparation of other f-element organometallic complexes.

## Methods

**General considerations**. See Supplementary Methods for further details.

**Ce(THF)$_2$(MBP)$_2$ (1)**. We previously reported a synthesis of **1** that could not be separated from the lithium halide byproducts[29]. This revised method provides clean **1**. In an N$_2$ filled drybox, to a clear, colorless solution of H$_2$MBP (0.270 g, 0.793 mmol, 2 equiv) in 4 mL of THF in a 20 mL scintillation vial with a Teflon coated stir bar, was added a yellow solution of Ce(O$^t$Bu)$_4$(THF)$_2$ (0.200 g, 0.396 mmol, 1 equiv) in a 6 mL solution of 2:1 THF:benzene at room temperature with stirring. The reaction immediately turned an intense purple color and was stirred for 1 h. The volatile materials were removed under reduced pressure, the residue was triturated with 2 mL of benzene to liberate the *tert*-butanol byproduct, and the volatile materials were again removed under reduced pressure. The resulting purple solid was transferred onto a medium porosity fritted filter and washed with 5 × 2 mL of pentane. The purple solid was dried under reduced pressure for 3 h. Yield: 0.311 g, 0.324 mmol, 82%.

NMR data for this complex was not previously reported and is provided here:
$^1$H NMR (400 MHz, THF-$d_8$) δ: 7.15 (s, 4H), 6.79 (s, 4H), 5.01 (d, $J$ = 13.4 Hz, 2H) 3.51 (d, $J$ = 14.0 Hz, 2H), 2.31 (s, 12 H), 1.44 (s, 36 H).

$^{13}$C{$^1$H} NMR (100 MHz, THF-$d_8$) δ: 168.12, 137.25, 134.45, 129.13, 128.17, 124.12, 35.40, 34.99, 31.15, 20.93.

The quantity of THF present for **1** was verified by $^1$H-NMR in C$_6$D$_6$.

**[Li(THF)][*ortho*-oxa] (2)**. Synthesis adapted from similar compounds[48]. In a N$_2$ filled drybox, a solution containing H-*ortho*-oxa (1.217 g, 5.0 mmol, 1 equiv) and 10 mL of hexanes in a 20 mL scintillation vial with a Teflon coated stir bar was

placed in a −30 °C freezer for 30 mins. The vial was removed from the freezer and, while stirring, a solution of *n*-butyl lithium (2.5 M, 5 mmol, 2 mL) was added dropwise over 5 min. The solution turned from colorless to yellow to brown and a yellow solid precipitated. The reaction mixture was stirred for 50 min at room temperature, after which the solid was collected by filtration over a coarse-porosity fritted-filter and subsequently washed with 3 × 2 mL of hexanes and 1 × 2 mL of pentane. The tan solid was then dried under reduced pressure for 2 h. The solid was then dissolved in minimal THF at rt and then placed in a −30 °C freezer overnight. Yellow crystalline blocks formed and were collected over a coarse-porosity fritted-filter and washed with 3 × 2 mL of pentane. The yellow blocks were dried for 2 h under reduced pressure. Yield: 0.831 g, 2.59 mmol, 52%.

$^1$H NMR (400 MHz, THF-$d_8$) δ: 8.28 (s, 1H), 7.57 (d, $J$ = 8.0 Hz, 1H), 7.07 (dd, $J$ = 8.0, 2.7 Hz, 1H), 4.17 (s, 2H), 1.34 (s, 6H).

$^{13}$C{$^1$H} NMR (101 MHz, THF-$d_8$) δ: 203.98, 172.88, 143.18 (q, $J$ = 2.2 Hz), 137.45 (q, $J$ = 3.3 Hz), 127.47 (q, $J$ = 274.1 Hz), 127.31 (q, $J$ = 28.2 Hz), 124.46, 119.28 (q, $J$ = 4.0 Hz), 80.39, 66.58, 29.03.

$^{19}$F NMR (376 MHz, THF-$d_8$) δ: −64.29

$^7$Li NMR (156 MHz, THF-$d_8$) δ: 2.08

Anal. Cal. for $C_{12}H_{11}F_3LiNO$•$(C_4H_8O)_{0.5}$: C, 58.96; H, 5.30; N, 4.91. Found C, 59.41; H, 5.41; N, 4.75.

The quantity of THF present for **2** was verified by $^1$H-NMR in $C_6D_6$.

**[Li(THF)$_4$][Ce(ortho-oxa)(MBP)$_2$ (3-THF)**. In an N$_2$ filled drybox, two 20 mL scintillation vials were placed in a −30 °C freezer. One contained a dark purple solution of **1** (0.200 g, 0.208 mmol, 1 equiv) in 4 mL of benzene with a Teflon coated stir bar and the other contained a yellow solution of **2** (0.067 g, 0.208 mmol, 1 equiv) in 4 mL of benzene. After cooling for 30 min, the now frozen solution of **2** was removed from the freezer and allowed to thaw. Immediately upon thawing, the frozen solution of **1** was removed from the freezer and the solution of **2** was added dropwise at rt over 2 min. Upon mixing, the solution immediately changed from a dark purple to a dark red color and was allowed to stir for 5 min at rt. At this point the volatile materials were removed under vacuum. The resulting solid was redissolved in a mixture of 3 mL of toluene and 8 drops of THF in an 8 mL scintillation vial. This solution was layered with 5 mL of pentane and placed in a −30 °C freezer for 3 days. During this time, red crystals formed, and were collected by filtration over a medium porosity fritted filter and washed with cold pentane 5 × 2 mL. Yield: 0.198 g, 0.137 mmol, 66%.

$^1$H NMR (500 MHz, THF-$d_8$) δ: 8.53 (s, 1H), 7.78 (d, $J$ = 8.0 Hz, 1H), 7.12 (d, $J$ = 7.5 Hz, 1H), 7.01 (s, 2H), 6.92 (s, 2H), 6.75 (s, 1H), 6.66 (s, 1H), 6.62 (s, 1H), 6.58 (s, 1H), 5.10 (d, $J$ = 13.4 Hz, 1H), 4.72 (d, $J$ = 13.5 Hz, 1H), 4.32 (s, 1H), 4.01 (s, 1H), 3.23 (d, $J$ = 13.4 Hz, 1H), 3.10 (d, $J$ = 13.6 Hz, 1H), 2.27 − 2.05 (m, 12H), 1.57 (s, 3H), 1.45 (s, 9H), 1.39 (s, 9H), 1.24 (s, 3H), 1.16 (s, 9H), 1.08 (s, 9H).

$^{13}$C{$^1$H} NMR (126 MHz, THF-$d_8$) δ: 255.58, 174.49, 168.21, 167.40, 166.81, 137.97, 137.72, 137.33, 137.11, 136.98, 136.79, 135.03, 134.90, 134.58, 132.58 (q, $J$ = 3.1 Hz), 131.11 (q, $J$ = 29.3 Hz), 128.85, 128.75, 128.61, 127.02, 126.64 (q, $J$ = 177 Hz), 126.08, 124.25, 124.11, 123.95, 123.60, 120.18 (q, $J$ = 4.1 Hz), 82.35, 68.88, 35.89, 35.71, 35.60, 35.03, 32.46, 31.54, 31.09, 30.97, 30.16, 21.31, 21.17.

$^{19}$F NMR (470 MHz, THF-$d_8$) δ: −62.44

$^7$Li NMR (194 MHz, THF-$d_8$) δ: −0.57

X-ray quality crystals were obtained from a vapor diffusion of pentane into concentrated solutions of 3 in a solution consisting of 1:2 THF:toluene in a −30 °C freezer.

Anal. Cal. for $C_{74}H_{99}CeF_3LiNO_9$•$(C_7H_8)$: C, 65.61; H, 7.66; F, 3.95; N, 1.03. Found C, 65.21; H, 6.65; N, 1.30. Best result of three attempts.

UV-Vis: λ = 460 nm (ε = 7533 Lmol$^{-1}$ cm$^{-1}$), λ = 292 nm (ε = 24,426 Lmol$^{-1}$ cm$^{-1}$).

**[Li(DME)$_3$][Ce(ortho-oxa)(MBP)$_2$ (3-DME)**. In an N$_2$ filled drybox, two 20 mL scintillation vials were placed in a − 30 °C freezer. One contained a dark purple solution of **1** (0.100 g, 0.104 mmol, 1 equiv) in 2 mL of benzene with a Teflon coated stir bar and the other contained a yellow solution of **2** (0.034 g, 0.104 mmol, 1 equiv) in 2 mL of benzene. After cooling for 30 min, the now frozen solution of **2** was removed from the freezer and allowed to thaw. Immediately upon thawing, the frozen solution of **1** was removed from the freezer and the solution of **2** was added dropwise at rt over 2 min. Upon mixing, the solution immediately changed from a dark purple to a dark red color and was allowed to stir for 5 min at rt. At this point the volatile materials were removed under reduced pressure. The resulting solid was dissolved in of 3 mL of DME in an 8 mL scintillation vial. This solution was layered with 5 mL of pentane and placed in a −30 °C freezer for 3 days. During this time, red crystals formed, and were collected by filtration over a medium porosity fritted filter and washed with cold pentane 5 × 2 mL. Yield: 0.111 g, 0.079 mmol, 75%.

$^1$H NMR (500 MHz, THF-$d_8$) δ: 8.53 (s, 1H), 7.80 (d, $J$ = 7.9 Hz, 1H), 7.13 (dd, $J$ = 8.1 Hz, 2.6 Hz, 1H), 7.01 (s, 2H), 6.91 (s, 2H), 6.76 (s, 1H), 6.68 (s, 1H), 6.64 (s, 1H), 6.59 (s, 1H), 5.12 (d, $J$ = 13.3 Hz, 1H), 4.72 (d, $J$ = 13.4 Hz, 1H), 4.33 (d, $J$ = 7.6 Hz, 1H), 4.01 (d, $J$ = 6.6 Hz 1H), 3.24 (d, $J$ = 13.6 Hz, 1H), 3.11 (d, $J$ =

13.6 Hz, 1H), 2.24 (s, 3H), 2.19 (s, 9H), 1.58 (s, 3H), 1.46 (s, 9H), 1.40 (s, 9H), 1.25 (s, 3H), 1.17 (s, 9H), 1.09 (s, 9H).

$^{13}$C{$^1$H} NMR (126 MHz, THF-$d_8$) δ: 255.61, 174.55, 168.23, 167.46, 166.87, 138.03, 137.77, 137.40, 137.18, 137.05, 136.84, 135.08, 134.94, 134.63, 132.65 (q, $J$ = 3.5 Hz), 131.18 (q, $J$ = 29.5 Hz), 129.03, 128.82, 128.66, 127.22 (q, $J$ = 177 Hz), 127.06, 126.14, 124.29, 124.18, 123.99, 123.66, 120.24 (q, $J$ = 4.0 Hz), 82.41, 68.93, 35.94, 35.76, 35.68, 35.09, 32.52, 31.59, 31.14, 31.03, 30.22, 21.36, 21.21.

$^{19}$F NMR (470 MHz, THF-$d_8$) δ: −62.44

$^7$Li NMR (194 MHz, THF-$d_8$) δ: −0.57

X-ray quality crystals were obtained from a layering of pentane on top of a saturated solution of 3-DME in DME (1:1, DME:Pentane).

Anal. Cal. for $C_{70}H_{101}CeF_3LiNO_{11}$: C, 62.90; H, 7.62; N, 1.05. Found C, 62.45; H, 7.32; N, 1.55.

## Data availability

Crystallographic data for the structures reported in this article have been deposited at the Cambridge Crystallographic Data Center (CCDC) under deposition nos. CCDC 1998883 (**3-THF**) and 2043597 (**3-DME**). These data can be obtained free of charge from the Cambridge Crystallographic Data Centre via www.ccdc.cam.ac.uk/data_request/cif. All other data supporting the findings of this study are available within the Article and its Supplementary Information and from the corresponding authors upon reasonable request.

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

## Acknowledgements

E.J.S., P.J.W., and G.B.P. gratefully acknowledge the U.S. National Science Foundation (CHE-1955724 to E.J.S., CHE-1902509 to P.J.W., and Graduate Research Fellowship Program NSF-GRFP to G.B.P.). We also thank the University of Pennsylvania for support of this work. D.C.S. and J.A. acknowledge support from the U.S. Department of Energy, Office of Basic Energy Sciences, Heavy Elements program, under grant DE-SC0001136. We thank the Center for Computational Research (CCR) at the University of Buffalo for providing computational resources.

## Author contributions

G.B.P. and E.J.S conceived this project. G.B.P. performed the synthesis, electrochemical, NMR, and UV-Vis experiments. M.R.G. and P.J.C. collected and solved the X-ray structures. G.B.P., P.J.W., and E.J.S. analyzed the experimental data. D.-C.S. and J.A. conducted the theoretical computations and analyzed the results. G.B.P., D.-C.S., J.A., P.J.W. and E.J.S. participated in drafting the paper. All authors discussed the results and contributed the preparation of the final paper.

## Competing interests

The authors declare no competing interests.
