## [Peer Review File · Nature Communications]

REVIEWER COMMENTS

Reviewer #1 (Remarks to the Author):

Organocerium complexes of Ce(IV) are rare due to Ce(IV) being a powerful oxidant and the reductive C-C bond coupling with accompanying reduction to Ce(III) is more likely. The Schelter group has had a long standing presence in cerium chemistry and, in this report, they detail the synthesis and characterization of a Ce(IV) complex with a Ce-C(aryl) bond. The reaction of a bis(phenol) ligand with Ce(OtBu)₄(THF)₂ forms the previously reported Ce(IV) starting material more effectively, followed by reaction with an oxazolide to make the 'ate' product. The Ce-C bond has a highly shifted ¹³C NMR resonance at 255-256 ppm, which is ~50 ppm from the lithium shift, and far removed from the normal aryl region. Electronic structure and NMR calculations are employed to examine the Ce-Aryl complex in more detail.

Minor stuff:

In the introduction, ¹³C NMR resonances have been correlated to electronic structure in a couple other cases: *Organometallics* 2017, 36, 4519 and *Organometallics* 2018, 37, 1884. These should be cited due to their relation to this manuscript. In addition, *Inorganics* 2015, 3, 589 describes the synthesis of Ce(IV) Cp derivatives and should be added to those in references 15-18.

On page 4, line 82, pentamethylcyclopentadiene should be dienyl since the anion is what is being referenced.

The heading Discussion should be Conclusion, right?

In the experimental, for the ¹H and ¹³C NMR data, the 8 in d⁸-THF is superscripted, but subscripted in ¹⁹F and ⁷Li NMR data. All should be subscripted.

The only issue I have with the manuscript is the skirting of the other Ce(IV)-carbon bonds with heteroatoms. NHCs - fine. As pointed out in Hayton/Hrobarik's (H/H) work (OM 2017 above), Th-NHCs have similar ¹³C NMR resonances to free NHCs, so they are expected to have downfield shifts. Also, in the H/H work (as well as OM 2018 above), thorium methanediide ligands adopt M-C bonds in a normal M-C sigma bond range (78 ppm), hence the heteroatoms did not really change much. But cerium methanediide compounds are odd with ¹³C NMR resonances ranging 324-343 ppm! So, in most cases, except Ce, methanediides adopt every day M-C sigma bonds, so simply stating that heteroatoms make them different and do not warrant comparison is not compelling, at least not to me. Something is odd about cerium and it could very well be related to the ¹³C NMR shift observed here. I simply implore the authors to think about why this is happening because, to me, it has to do with cerium, not the ligand.

Overall, the rarity of Ce(IV) organometallic compounds makes this manuscript compelling, and brings about more questions for the f element community to debate. I support its publication in *Nat. Commun.* once comments/suggestions/concerns are addressed.

Reviewer #2 (Remarks to the Author):

This manuscript details the synthesis and characterization of an interesting and novel Ce⁴⁺ organometallic compound and provides an in-depth investigation via NMR, X-ray diffraction, and computational models. The authors state that NMR spectroscopy finds evidence of spin-orbit coupling in Ce⁴⁺ for the first time, and computational models reproduce the observed chemical shift. The single crystal X-ray diffraction (SCXRD) determined crystal structure is then used as a confirmation of appropriate bond lengths and angles. The authors present their findings well and succinctly, and while I cannot expertly comment on the NMR spectroscopy, I do have some reservations about the SCXRD collection and data work-up.

Some might argue that the SCXRD data is not necessary, due to the in-depth NMR and accompanying computation models, yet I find that additional confirmation (e.g. bond distances)

comforting when presenting new, unusual, or game-changing research. In this regard, the SCXRD has some issues that should not be ignored and should at least be commented on in the Supporting Information.

I believe an insufficient absorption correction was applied (or maybe unaccounted twinning). The authors, within the CheckCIF, state that there is "minor Ce atom disorder", and thus two-component ligand (or solvent) disorder cannot be effectively modeled. Heavy metals often have significant residual electron density and can sometimes be handled with face indexing of the crystal. I do not see this as "Ce atom disorder", which would imply that accurate bond distances between Ce and other atoms would be difficult to arrive at. Using a Mo or Ag source could also help with this absorption issue.

Additionally, there is an ISOR command present in the CIF and the authors have not provided which atom that the command was used on in the manuscript or SI (and the .res file is not imbedded in the CIF either). Briefly looking in the CIF, it seems that Ce1 might present as NPD – and could be what the ISOR is used on – which should be refined in a more appropriate (and "less-hard") way. The identity of the ISOR should be provided regardless, along with a brief paragraph describing the data/refinement work-up to explain what was done to the data (as there seems to be a significant number of site occupancy changes for solvent and ligand moieties).

I view these issues as quite minor when considering the scope of the manuscript (and believe the overall structure as presented), yet specific refinement treatment for the residual density, disorder, and use of ISOR should be investigated, applied, and provided before resubmission. Bond distances and angles may change slightly, though (hopefully) within error.

Reviewer #3 (Remarks to the Author):

Please see attached file with formatting.

Reviewer #4 (Remarks to the Author):

The authors report a very rare example of a complex presenting a Ce(IV)-C bond and the first Ce(IV)-aryl complex by using an original synthetic approach.

This finding is of high interest to the community and should be published in Nature Communications.

However, it is not evident to me why the authors define the ¹³C carbon shift at 255 ppm anomalous since it falls in the range of reported Ce(IV)-C shifts ranging from 213 to 343 ppm. Please clarify this aspect in the manuscript.

Moreover, the computational studies fail overall to fully distinguish if the carbon shift is due to covalency or to the presence of multiconfigurational states.

I would suggest to perform additional spectroscopic studies (VT magnetic studies or Xanes) to further assess the presence of multiconfigurational studies that would make harder to pinpoint the effects of covalency on the ipso carbon chemical shifts. It would be helpful to further elucidate how the carbon shift can be related to covalency if it results from a mix of electronic configurations.

Minor points

Page 6 line 107 : please provide a reference when mentioning the E_{1/2} value of the Ce(III)/Ce(IV) couple for 1

Page 7 line 109 give reference when mentioning reported CV of 2

The authors report the first Ce-C_{aryl} bonded complex, [Li(THF)₄][Ce^{IV}(κ²-ortho-oxa)(MBP)₂] and interrogate its bonding interactions through NMR spectroscopy and DFT calculations. The complex's uniquely downfield shifted C_{aryl} ¹³C{¹H} NMR shift and calculations suggest a significantly covalent Ce-C_{aryl} interaction and that the downfield shift is driven by the significant SOC of Ce^{IV}. The NMR work presented in this study is fantastic.

The synthetic chemistry, NMR spectroscopy, and non-trivial computational analysis are important benchmarks for the field, and certainly will generate significant excitement. However, there are a few key issues that must be addressed to secure these claims.

While I agree that the interrogation of multiconfigurational behavior in high-valent lanthanides is potentially an important aspect of this work, none of the data presented here, including the computational analysis, addresses this aspect and these conclusions are not supported.

I would suggest that if the key comments are addressed, this paper would be appropriate for a high-profile field journal (JACS, ACIE, Chem. Sci.). Since Nature Communications is committed to publishing results of significance to specialists within each broad field, Nature Communications would be an appropriate forum for a revised version of this manuscript.

This paper would be substantially more impactful if it reported experimental data that could address multiconfigurational behavior in **3**. However, this would require significant further experimentation including synchrotron studies, magnetometry, and/or CASSCF calculations. I would suggest that the synthetic and theoretical efforts described here be published as one complete story and the further necessary spectroscopic, magnetometry, and theoretical studies be published separately. These concerns are described below.

Key comments:

- 1) The authors state their hypothesis for the stabilization of a Ce-C_{aryl} bond as “Considering strategies to stabilize a Ce⁴⁺-C_{aryl} bond, we hypothesized that tethering the aryl group to the Ce center would kinetically inhibit homolysis of the Ce-C bond.”
 - a. This is quite reasonable but leaves unstated the other significant contribution to the redox stability of this system – the supporting aryloxy ligand framework which significantly stabilizes the tetravalent cerium center. In fact, the authors have demonstrated this stabilization in their prior work. This should also be clearly stated here, and, at a minimum, a few references to the use aryloxy ligand frameworks for the stabilization of reactive high-valent, organometallic early transition metal complexes should be included. I realize this is a bit of rabbit-hole, but the complexes and reactivity reported here have significant similarity to the work of Ian Rothwell and Hiroyuki Kawaguchi.
- 2) There is clearly unmodeled residual electron density in the structural model of the single-crystal diffraction data of **3**. Given that this complex is the centerpiece of this study, I would have liked more attention paid to this issue beyond the annotated CIF file. The authors claim that this unmodeled electron density is due to incompletely modeled whole molecule disorder in which only the Ce atoms of the two independent molecules in the asymmetric unit for the minor component can be identified.

- a. This may be the best model possible, however, the authors should include further details on the refinement in the SI about how this model was determined and what other models were tested. For example, what about lower symmetry with twinning? Or a modulated supercell? I would suggest that the authors also provide the SCXRD derived precession images in the SI in order to rule out the presence of a supercell or undiagnosed twinning.
 - b. Ideally, a non-problematic crystal would be identified. Have the authors tried to resolve the issues with this structure chemically? The supporting Li cation is supported by four THFs. These could be readily substituted by DME, TMEDA, or 12-c-4 in order to modify the crystal in order to get higher quality data. The authors could also consider metathesis with PPN or PPh₄ cations or their derivatives to help template the structure.
- 3) The authors provide no proof-of-purity for any of the new complexes reported in this study. This omission makes the reported yields somewhat unreliable. In the case of organometallic complexes, elemental analyses (EA, CHN) are standard. I expect that it is probably difficult to obtain EAs that match the expected values for several of these complexes. However, the discrepancies could be reasonably explained. The authors must report their EA values for their complexes.
- 4) There is some confusion introduced in the text with the authors' inconsistent use of mixed-valent and multiconfigurational. Additionally, the relationship of these phenomena to canonical oxidation states is not presented in a clear and consistent manner. The confusion or conflation of terms is common in the literature. This paper would benefit from clarification of how these authors are using the terms. I would suggest that mixed-valent is best used to 1) refer to complexes, co-crystals, or materials in which the same metal-ion exists in two or more canonical oxidation states, or 2) to refer to a complex of an f-element or a main group element that has a ground state electronic structure that has mixed-quantum numbers (e.g. 4fⁿ5d¹). None of the complexes in this study (nor any of the tetravalent Ce complexes cited by the authors) fall into either definition. All tetravalent Ce complexes and materials that have been interrogated by L₃-edge XANES to-date can be described as multiconfigurational (by the definition in refs 6 and 33 in this manuscript). While there are several new inorganic tetravalent cerium complexes that present challenges for the current understanding of the phenomenon, the dichotomy presented in line 143-144, "common feature for CeIV compounds as well as for mixed-valence CeIV/CeIII species," has not been demonstrated to exist.
 - a. The authors present a Ce configuration retrieved from NBO analysis for **3** (4f^{0.76}d^{0.60}) as evidence for the partial *f* character in **3** and imply, without any comment or support, that this *f* character (0.76) is an equivalent observable to that extracted from the fit of L₃-edge XANES spectra of tetravalent cerium materials and complexes (the authors choose the *n_f* values of CeO₂ (0.58) and cerocene (0.80) for comparison).
 - i. If this DFT derived value corresponds to the *n_f* values derived from L₃-XANES spectra, then it would be a landmark result. Several groups have tried to construct a theoretical model of multiconfigurational behaviour that is correlated with experimental observables, and have not yet been successful.

1. However, in the absence of experimental data such as transmission L₃-edge XANES spectra, HERFD XANES spectra, or even dc susceptibility for **3**, this analysis of the NBO results is unsubstantiated and should be removed or softened (specifically the two sentences in lines 170-174).
 - a. I would expect the NBO derived configuration to be sensitive to method (not unlike the calculated NMR shift). The development of this analysis must be paired with experimental observables.
 - b. The proposed d^{0.60} character goes undescribed – this presents significant challenge theoretically and experimentally. What would the comparable CASSCF calculations show? dc susceptibility and RIXS experiments would be informative for future studies.
- b. To be clear, I am excited that the authors are trying to make this connection. However, I believe substantial evidence required to establish this model and would need to be addressed in a separate manuscript on the development and analysis of this theoretical model and its correspondence to the reported L₃-edge XAS data in the literature.

Minor points:

Line 38: “cyclopentadieninde” should be “cyclopentadienide”

Line 96: This should be E_{pc} not E_{pa} : “The E_{pa} of **3**, -1.67 V versus Fc/Fc+, shifts by -0.72 V relative to the E1/2 of **1** (-0.94 V vs. Fc/Fc+), indicating that the ortho-oxa- moiety significantly stabilizes the CeIV couple in THF.”

Line 98: E_{pc} should be E_{pa} here since you say oxidation: The reduction of **3** is not reversible under the electrochemical conditions, although the event precedes a reversible oxidation at E1/2 = -0.94 V versus Fc/Fc+ and an irreversible oxidation at E_{pc} = -0.43 V.

Line 109: Again, “E_{pc}” should be “E_{pa}” since it is an anodic scan. “Indeed, the return anodic scan comprises waves at E1/2 = -0.94 V versus Fc/Fc+ and E_{pc} = -0.43 V respectively, consistent with previous assignments for compounds **1** and **2** (Figure 4).”

Line 110: An event at -0.43 V is not actually shown in Figure 4 as mentioned in line 110. Has this E_{pc} value been reported in error? Or is it not shown and the reader should be directed to an SI figure such as S4b where there does seem to be an E_{pa} around -0.5 V? Additionally, if these are from previous assignments as stated, please add references to the end of this sentence.

Line 124: “¹³C{1H} shift at ~213 ppm for a the CeIV-NHC” remove “a”

Line 155: I believe there should be a “two” between “are” and “two-center” “There are two-center two-electron σ bonds describing the donation bonding between the aryl carbon and oxazolinide nitrogen and Ce”

Is Fig. 3 an ORTEP or is it really a Mercury plot with a partial thermal ellipsoids? Either way, the crystallographic section of the SI or the main text should have the appropriate reference for the graphic generation.

Fig. 4 should have the scan rate listed.

Values in Figure 4 caption and two separate places in the text (line 100 and 109) text do not match. -0.94 vs -0.95 V. Please clarify the correct value or if these are in reference to different scan rates.

Responses to the Reviewer Comments (NCOMMS-20-25277;)

Note, changes in response to the reviewers are highlighted with blue text.

Reviewer #1 (Remarks to the Author):

Organocerium complexes of Ce(IV) are rare due to Ce(IV) being a powerful oxidant and the reductive C-C bond coupling with accompanying reduction to Ce(III) is more likely. The Schelter group has had a long standing presence in cerium chemistry and, in this report, they detail the synthesis and characterization of a Ce(IV) complex with a Ce-C(aryl) bond. The reaction of a bis(phenol) ligand with Ce(OtBu)₄(THF)₂ forms the previously reported Ce(IV) starting material more effectively, followed by reaction with an oxazolide to make the 'ate' product. The Ce-C bond has a highly shifted ¹³C NMR resonance at 255-256 ppm, which is ~50 ppm from the lithium shift, and far removed from the normal aryl region. Electronic structure and NMR calculations are employed to examine the Ce-Aryl complex in more detail.

Minor stuff:

In the introduction, ¹³C NMR resonances have been correlated to electronic structure in a couple other cases: Organometallics 2017, 36, 4519 and Organometallics 2018, 37, 1884. These should be cited due to their relation to this manuscript. In addition, Inorganics 2015, 3, 589 describes the synthesis of Ce(IV) Cp derivatives and should be added to those in references 15-18.

Response: The three requested references have been added.

On page 4, line 82, pentamethylcyclopentadiene should be dienyl since the anion is what is being referenced.

Response: “pentamethylcyclopentadiene” has been changed to “pentamethylcyclopentadienyl”

The heading Discussion should be Conclusion, right?

Response: Previous Nature Communication articles have used “Discussion” rather than “Conclusion” as shown here: <https://www.nature.com/articles/s41467-020-17990-z.pdf> and <https://www.nature.com/articles/s41467-020-18041-3.pdf> So, we retain the “Discussion” section heading.

In the experimental, for the ¹H and ¹³C NMR data, the 8 in d₈-THF is superscripted, but subscripted in ¹⁹F and ⁷Li NMR data. All should be subscripted.

Response: This issue has been corrected in both the body of the manuscript and the Supporting Information.

The only issue I have with the manuscript is the skirting of the other Ce(IV)-carbon bonds with heteroatoms. NHCs - fine. As pointed out in Hayton/Hrobarik's (H/H) work (OM 2017 above), Th-NHCs have similar ¹³C NMR resonances to free NHCs, so they are expected to have downfield shifts. Also, in the H/H work (as well as OM 2018 above), thorium methanediide ligands adopt M-C bonds in a normal M-C sigma bond range (78 ppm), hence the heteroatoms

did not really change much. But cerium methanediide compounds are odd with ^{13}C NMR resonances ranging 324-343 ppm! So, in most cases, except Ce, methanediides adopt every day M-C sigma bonds, so simply stating that heteroatoms make them different and do not warrant comparison is not compelling, at least not to me. Something is odd about cerium and it could very well be related to the ^{13}C NMR shift observed here. I simply implore the authors to think about why this is happening because, to me, it has to do with cerium, not the ligand.

Response: While the reviewer is correct in noting the exceptional ^{13}C resonance for methanediide ligands bound to cerium, the statement that the cerium has little to do with the associated shift compared to the ligand, we feel is diminishing of the impact the alpha-phosphorus atoms have on the interaction of the carbon atom. Referencing the Hayton/Hrobarik OM 2017 paper, the SOC effects from the interaction between the thorium and carbon atom, the methanediide ligand has a SOC effect of only 3 ppm, but simply removing one phosphorous increases the SOC effect increases by 50 ppm, so there is a big impact of the ortho-phosphorus. Unfortunately, there is no isolated analogous cerium compound to see if the SOC effects are analogous between cerium and thorium, nor have the methanediide-Ce SOC effects been evaluated. We hope to make more connections between thorium and cerium chemical shifts in future publications.

Overall, the rarity of Ce(IV) organometallic compounds makes this manuscript compelling, and brings about more questions for the f element community to debate. I support its publication in Nat. Commun. once comments/suggestions/concerns are addressed.

We thank the reviewer for their supportive assessment of our work.

Reviewer #2 (Remarks to the Author):

This manuscript details the synthesis and characterization of an interesting and novel Ce $^{4+}$ organometallic compound and provides an in-depth investigation via NMR, X-ray diffraction, and computational models. The authors state that NMR spectroscopy finds evidence of spin-orbit coupling in Ce $^{4+}$ for the first time, and computational models reproduce the observed chemical shift. The single crystal X-ray diffraction (SCXRD) determined crystal structure is then used as a confirmation of appropriate bond lengths and angles. The authors present their findings well and succinctly, and while I cannot expertly comment on the NMR spectroscopy, I do have some reservations about the SCXRD collection and data work-up.

Some might argue that the SCXRD data is not necessary, due to the in-depth NMR and accompanying computation models, yet I find that additional confirmation (e.g. bond distances) comforting when presenting new, unusual, or game-changing research. In this regard, the SCXRD has some issues that should not be ignored and should at least be commented on in the Supporting Information.

I believe an insufficient absorption correction was applied (or maybe unaccounted twinning).

Response: We have responded to these comments on the x-ray structural data for (previously **3** now) **3-THF** individually, below. Additionally, for this revision, a second data set, comprising the DME solvate of the Li⁺ cation instead of the THF-solvate, complex **3-DME** in the revised version, was obtained. The bond parameters in the title complex are nearly identical to the new one, and the new data set does not have these same issues with the single crystal x-ray data collected for **3-THF**.

In the original structure of **3-THF** that we presented, we did attempt to find a twin component, with no success. Indexing fit 87% of all peaks harvested (150000) and no reasonable twin matrix was found using CrysAlisPro that fits the missed peaks.

The authors, within the CheckCIF, state that there is “minor Ce atom disorder”, and thus two-component ligand (or solvent) disorder cannot be effectively modeled. Heavy metals often have significant residual electron density and can sometimes be handled with face indexing of the crystal. I do not see this as “Ce atom disorder”, which would imply that accurate bond distances between Ce and other atoms would be difficult to arrive at. Using a Mo or Ag source could also help with this absorption issue.

Response: The use of face indexing and altering absorption correction values/variables did not resolve the issue of the large Q-peaks. When the Q-peaks are modeled as disordered Ce atoms, we get a 10% and 8% occupancy respectively. And the R factor drops to around 8.7%. We can see some evidence of a whole molecule disorder from Q-peaks that slightly resemble phenyl rings that correspond to the ligand, but the Q-peaks are too small (less than 0.8) and the amount of restraints/constraints needed would not benefit this structure and finding the entire second component of the light atoms is not plausible without using harsh constraints like SAME or EADP. In addition, the disordered component might be an enantiomer and that would make modeling it even more challenging without enough Q-peaks to find the orientation/position of the ligands. This structure was collected on Cu radiation due to the crystal being weakly diffracting with Mo and the long cell axes.

Additionally, there is an ISOR command present in the CIF and the authors have not provided which atom that the command was used on in the manuscript or SI (and the .res file is not imbedded in the CIF either). Briefly looking in the CIF, it seems that Ce1 might present as NPD...

Response: Ce1 is not NPD and has no restraints/constraints in our model.

– and could be what the ISOR is used on – which should be refined in a more appropriate (and “less-hard”) way. The identity of the ISOR should be provided regardless, along with a brief paragraph describing the data/refinement work-up to explain what was done to the data (as there seems to be a significant number of site occupancy changes for solvent and ligand moieties).

Response: ISOR and SIMU are used on the disordered THFs on the Li adduct, toluene solvent and bisphenol ligand with stretchy ellipsoids due to the presence of a whole molecule disorder.

I view these issues as quite minor when considering the scope of the manuscript (and believe the overall structure as presented), yet specific refinement treatment for the residual density,

disorder, and use of ISOR should be investigated, applied, and provided before resubmission. Bond distances and angles may change slightly, though (hopefully) within error.

Response: Comparing the bond lengths with the Ce disorder modeled and without shows no significant change in bond distances, all changes within error.

To reiterate, a second structure, the DME solvate, complex **3-DME**, was obtained and the bond parameters are very similar and it does not have these same issues with the SCXRD data of **3-THF**.

We appreciate the reviewer's careful analysis of our SCXR data.

Reviewer 3:

The authors report the first Ce-C_{aryl} bonded complex, [Li(THF)₄][Ce^{IV}(κ²-ortho-oxa)(MBP)₂] and interrogate its bonding interactions through NMR spectroscopy and DFT calculations. The complex's uniquely downfield shifted C_{aryl} ¹³C {¹H} NMR shift and calculations suggest a significantly covalent Ce-C_{aryl} interaction and that the downfield shift is driven by the significant SOC of Ce^{IV}. The NMR work presented in this study is fantastic.

Response: We thank the reviewer for their enthusiastic response.

The synthetic chemistry, NMR spectroscopy, and non-trivial computational analysis are important benchmarks for the field, and certainly will generate significant excitement. However, there are a few key issues that must be addressed to secure these claims.

While I agree that the interrogation of multiconfigurational behavior in high-valent lanthanides is potentially an important aspect of this work, none of the data presented here, including the computational analysis, addresses this aspect and these conclusions are not supported.

I would suggest that if the key comments are addressed, this paper would be appropriate for a high-profile field journal (JACS, ACIE, Chem. Sci.). Since Nature Communications is committed to publishing results of significance to specialists within each broad field, Nature Communications would be an appropriate forum for a revised version of this manuscript. This paper would be substantially more impactful if it reported experimental data that could address multiconfigurational behavior in **3**. However, this would require significant further experimentation including synchrotron studies, magnetometry, and/or CASSCF calculations. I would suggest that the synthetic and theoretical efforts described here be published as one complete story and the further necessary spectroscopic, magnetometry, and theoretical studies be published separately. These concerns are described below.

Response: We agree with the reviewer that there are further electronic structure studies to perform on this system. As a practical matter, it is not currently possible for us to collect the synchrotron data for such a study, due to COVID related restrictions of measurements at our preferred synchrotron, SSRL. As indicated by the reviewer, we prefer to report that work in a

subsequent publication. We have initiated that work, however, and the results will be presented in due course.

Key comments:

1) The authors state their hypothesis for the stabilization of a Ce-C_{aryl} bond as “Considering strategies to stabilize a Ce⁴⁺-C_{aryl} bond, we hypothesized that tethering the aryl group to the Ce center would kinetically inhibit homolysis of the Ce-C bond.”

a. This is quite reasonable but leaves unstated the other significant contribution to the redox stability of this system – the supporting aryloxy ligand framework which significantly stabilizes the tetravalent cerium center. In fact, the authors have demonstrated this stabilization in their prior work. This should also be clearly stated here, and, at a minimum, a few references to the use aryloxy ligand frameworks for the stabilization of reactive high-valent, organometallic early transition metal complexes should be included. I realize this is a bit of rabbit-hole, but the complexes and reactivity reported here have significant similarity to the work of Ian Rothwell and Hiroyuki Kawaguchi.

Response: we agree with the reviewer and have added the statements: “Lastly, we sought a supporting ligand that would stabilize the Ce^{IV} oxidation state to prevent charge transfer and subsequent Ce-C bond homolysis.” And “Aryloxy ligands have been previously shown to both stabilize the Ce^{IV} oxidation state and other high valent organometallic species.” Which includes several references from the above listed authors.

2) There is clearly unmodeled residual electron density in the structural model of the single crystal diffraction data of **3**. Given that this complex is the centerpiece of this study, I would have liked more attention paid to this issue beyond the annotated CIF file. The authors claim that this unmodeled electron density is due to incompletely modeled whole molecule disorder in which only the Ce atoms of the two independent molecules in the asymmetric unit for the minor component can be identified.

a. This may be the best model possible, however, the authors should include further details on the refinement in the SI about how this model was determined and what other models were tested. For example, what about lower symmetry with twinning? Or a modulated supercell? I would suggest that the authors also provide the SCXRD derived precession images in the SI in order to rule out the presence of a supercell or undiagnosed twinning.

Response: Thank you, reviewer, for the careful consideration of the crystal structure data. For the data set for **3-THF**, we were not able to determine any reasonable twinning model (using CrysAlisPro or Platon). The chosen cell indexed as 86.7% to 150000 reflections and no reliable twin orientation was able to cover the remaining 13% (see our response to the previous reviewer comment above). In addition, the histograms show no evidence of weak satellite peaks that would indicate a modulated cell. In our experience with modulated cells, multiple cell sizes can be indexed and weak in-between peaks can be identified, along with heavy disorder in several molecules in the asymmetric unit cell. No smaller cells can be identified, even by restricting cell

lengths. In addition, the precession images look fine with no indication of twinning or modulation.

b. Ideally, a non-problematic crystal would be identified. Have the authors tried to resolve the issues with this structure chemically? The supporting Li cation is supported by four THFs. These could be readily substituted by DME, TMEDA, or 12-c-4 in order to modify the crystal in order to get higher quality data. The authors could also consider metathesis with PPN or PPh₄ cations or their derivatives to help template the structure.

Response: Based on the reviewer's suggestion, we have investigated the recrystallization of the title compound from DME. Indeed, we have found that the recrystallization of the compound from DME afforded formation of the Li(DME)₃ cation, which produced a much higher-quality crystal overall. We have fully characterized this new congener of the compound, **3-DME**, and added it to the manuscript. This new congener has afforded better determination of important bond distances. We thank the reviewer for their suggestion.

3) The authors provide no proof-of-purity for any of the new complexes reported in this study. This omission makes the reported yields somewhat unreliable. In the case of organometallic complexes, elemental analyses (EA, CHN) are standard. I expect that it is probably difficult to obtain EAs that match the expected values for several of these complexes. However, the discrepancies could be reasonably explained. The authors must report their EA values for their complexes.

Response: We agree with the reviewer and typically obtain and report these data as a matter of course. Our initial submission was limited in this manner by COVID-19-related suspension of research activities. We have now provided EA data for compounds **2**, **3-THF**, and **3-DME**. We appreciate the reminder from the reviewer on this point.

4) There is some confusion introduced in the text with the authors' inconsistent use of mixed-valent and multiconfigurational. Additionally, the relationship of these phenomena to canonical oxidation states is not presented in a clear and consistent manner. The confusion or conflation of terms is common in the literature. This paper would benefit from clarification of how these authors are using the terms. I would suggest that mixed-valent is best used to 1) refer to complexes, co-crystals, or materials in which the same metal-ion exists in two or more canonical oxidation states, or 2) to refer to a complex of an f-element or a main group element that has a ground state electronic structure that has mixed-quantum numbers (e.g. 4fⁿ 5d¹). None of the complexes in this study (nor any of the tetravalent Ce complexes cited by the authors) fall into either definition. All tetravalent Ce complexes and materials that have been interrogated by L₃-edge XANES to-date can be described as multiconfigurational (by the definition in refs 6 and 33 in this manuscript). While there are several new inorganic tetravalent cerium complexes that present challenges for the current understanding of the phenomenon, the dichotomy presented in line 143-144, "common feature for CeIV compounds as well as for mixed-valence CeIV/CeIII species," has not been demonstrated to exist.

Response: We agree with the reviewer confusion exists in the literature wherein terms: "mixed-valent" and "multiconfigurational" are used (erroneously) interchangeably. And we agree with the

reviewer on the definition of a mixed-valent state, and with the fact that its meaning differs from that of a multiconfigurational state. To clarify the text, we have removed all occurrences of “mixed-valent” where the intention was to describe a system as “multiconfigurational”. Along the same lines, we changed the text: “The MOs with the most Ce 4f character remain largely metal-centered and span the seven lowest unoccupied molecular orbitals (LUMO to LUMO+6, **Supplementary Figures 11-17**) of the complex, a common feature for Ce^{IV} compounds as well as for mixed-valence Ce^{IV}/Ce^{III} species.” to “The MOs with the most Ce 4f character remain largely metal-centered and span the seven lowest unoccupied molecular orbitals (LUMO to LUMO+6, **Supplementary Figures 13-19**) of the complex, a common feature for Ce^{IV} compounds as well as for cerium species with a debated Ce^{IV}/Ce^{III} oxidation state.”

a. The authors present a Ce configuration retrieved from NBO analysis for **3** ($4f^{0.76}d^{0.60}$) as evidence for the partial f character in **3** and imply, without any comment or support, that this f character (0.76) is an equivalent observable to that extracted from the fit of L₃-edge XANES spectra of tetravalent cerium materials and complexes (the authors choose the n_f values of CeO₂ (0.58) and cerocene (0.80) for comparison).

Response: The Ce natural configuration of $4f^{0.76}5d^{0.60}$ is a result of donation bonding involving cerium and the neighboring C, N and O atoms, as evidenced by the NLMO analysis presented in Table 1 of the manuscript. This configuration is indeed similar to that in CeO₂ ($4f^{0.82}5d^{1.05}$) calculated by Hay et al. [J. Chem. Phys. 125, 034712, 2006] and to that in cerocene ($4f^{0.98}5d^{1.36}$) calculated by Moosen and Dolg [Chem. Phys. Lett. 594, 47-50, 2014], obtained via Mulliken population analyses of calculated ground state electron densities. Indeed, for both CeO₂ and cerocene, the calculated Ce configuration agrees well with the configuration derived from the L₃-edge XAS ($4f^{0.58}$ for CeO₂ and $4f^{0.89}$ for cerocene, values quoted as “ n_f ” in the XAS community – the combined 4f shell occupation), as far as the 4f occupancy is concerned. We did not mean to say that the calculated Ce configuration for complex **3** is strictly equivalent to those derived experimentally for CeO₂ and cerocene. To resolve this issue, the original sentence at lines 170, 171, now reads: “The large Ce 4f electron count of **3** (0.76), associated mainly with the sizable Ce-C_{aryl} bonding, is comparable to the calculated and experimentally-determined Ce 4f electron counts in CeO₂ and Ce(C₈H₈)₂.”

i. If this DFT derived value corresponds to the n_f values derived from L₃-XANES spectra, then it would be a landmark result. Several groups have tried to construct a theoretical model of multiconfigurational behaviour that is correlated with experimental observables, and have not yet been successful.

Response: The n_f value derived from peak-fitting an experimental L₃ XAS spectrum is believed to give a measure of the 4f-shell population in the GS of the complex. Then, yes, our DFT/NBO-derived Ce $4f^{0.76}$ occupation, which is in fact the combined 4f-shell electron occupation, can be taken as a calculated analogue of n_f , with the caveat that its accuracy connects to the chemical bonding in the complex as captured by the used approximation, i.e. DFT/B3LYP. However, in absence of the L₃ XAS experiment we prefer not to overinterpret the calculations, and we refrain from identifying the DFT-calculated cerium 4f population of 0.76 with n_f for the purpose of the

present manuscript. CASSCF calculations of n_f and of the L_3 XAS spectrum are under way and will be presented in a separate work.

1. However, in the absence of experimental data such as transmission L3-edge XANES spectra, HERFD XANES spectra, or even dc susceptibility for **3**, this analysis of the NBO results is unsubstantiated and should be removed or softened (specifically the two sentences in lines 170-174).

Response: The sentences in lines 170-174 have been changed from “The large Ce 4f electron count of **3** (0.76) is bracketed by the Ce 4f electron counts in CeO₂ (0.58) and in Ce(COT)₂ (0.82).⁴¹ This may indicate a multi-configurational Ce^{III/IV} character, as commonly assigned for cerocene, which manifests itself in the DFT calculations as a pronounced donation bonding.” to “The large Ce 4f electron count of **3** (0.76), associated mainly with the sizable Ce-C_{aryl} bonding, is comparable to the calculated and experimentally-determined Ce 4f electron counts in CeO₂ and Ce(C₈H₈)₂.⁴¹⁻⁴³ We anticipate that this similarity has important implications regarding the electronic structure of **3**, in the sense that it may potentially exhibit a multi-configurational ground-state wavefunction with Ce^{III/IV} character, similar to cerocene. However, further spectroscopic studies are needed, and are under way, to confirm this assignment for **3**. ”

- a. I would expect the NBO derived configuration to be sensitive to method (not unlike the calculated NMR shift). The development of this analysis must be paired with experimental observables.

Response: In the revised manuscript, we provide electronic structure details derived from the same DFT/PBEh-40 calculation that gave the NMR shift most similar to experiment. The Ce-C_{aryl} chemical bonding is very similar to that obtained with B3LYP, i.e. there is significant f-character in the Ce-C_{aryl} bond. The Ce natural electron configuration is 4f^{0.58}5d^{0.59} with PBEh-40, showing the sizable donation bonding into the Ce 4f and 5d shells similar to B3LYP. It is worth noting that a 4f^{0.58} is also shared by CeO₂, a system (material) allegedly iconic for cerium tetravalency. Similar are also Ce configurations predicted by Mulliken population analyses: 4f^{0.90}5d^{1.09} with DFT/B3LYP, 4f^{0.74}5d^{0.99} with DFT/PBEh-40, 4f^{0.86}5d^{1.08} with DFT/PBE0. I.e. all these methods stand for sizable ligand-to-metal donation bonding, which involves both the Ce 4f and 5d atomic orbitals.

- b. The proposed d^{0.60} character goes undescribed – this presents significant challenge theoretically and experimentally. What would the comparable CASSCF calculations show? dc susceptibility and RIXS experiments would be informative for future studies.

Response: D-character for Ce^{IV}-X bonds is quite common, we do plan on investigating this compound further using some of the methods mentioned here in a future publication. See also the answer to comment 4.a, above.

b. To be clear, I am excited that the authors are trying to make this connection. However, I believe substantial evidence required to establish this model and would need to be addressed in a separate manuscript on the development and analysis of this theoretical model and its correspondence to the reported L₃-edge XAS data in the literature.

Response: Indeed, further work on this system is underway, as indicated, and will be reported in due course.

Minor points:

Line 38: “cyclopentadieninde” should be “cyclopentadienide”

Response: “Ce^{IV} complexes of cyclopentadieninde” has been changed to “Ce^{IV} complexes of cyclopentadienide”

Line 96: This should be E_{pc} not E_{pa}: “The E_{pa} of **3**, -1.67 V versus Fc/Fc+, shifts by -0.72 V relative to the E_{1/2} of **1** (-0.94 V vs. Fc/Fc+), indicating that the ortho-oxa- moiety significantly stabilizes the Ce^{IV} couple in THF.”

Response: “The E_{pa} of **3**” has been changed to “The E_{pc} of **3-THF**”

Line 98: E_{pc} should be E_{pa} here since you say oxidation: The reduction of **3** is not reversible under the electrochemical conditions, although the event precedes a reversible oxidation at E_{1/2} = -0.94 V versus Fc/Fc⁺ and an irreversible oxidation at E_{pc} = -0.43 V.

Response: “an irreversible oxidation at E_{pc}” has been changed to “an irreversible oxidation at E_{pa}”

Line 109: Again, “E_{pc}” should be “E_{pa}” since it is an anodic scan. “Indeed, the return anodic scan comprises waves at E_{1/2} = -0.94 V versus Fc/Fc+ and E_{pc} = -0.43 V respectively, consistent with previous assignments for compounds **1** and **2** (Figure 4).”

Response: “and E_{pc} = -0.43 V” has been changed to “and E_{pa} = -0.43 V”

Line 110: An event at -0.43 V is not actually shown in Figure 4 as mentioned in line 110. Has this E_{pc} value been reported in error? Or is it not shown and the reader should be directed to an SI figure such as S4b where there does seem to be an E_{pa} around -0.5 V? Additionally, if these are from previous assignments as stated, please add references to the end of this sentence.

Response: The reference to Supplemental Figures 4a and 4b have been added.

Line 124: “¹³C{¹H} shift at ~213 ppm for a the Ce^{IV}-NHC” remove “a”

Response: “¹³C{¹H} shift at ~213 ppm for a the Ce^{IV}-NHC” has been changed to “¹³C{¹H} shift at ~213 ppm for the Ce^{IV}-NHC”

Line 155: I believe there should be a “two” between “are” and “two-center” “There are two-center two-electron σ bonds describing the donation bonding between the aryl carbon and oxazolinide nitrogen and Ce”

Response: “There are two-center two-electron σ bonds” has been changed to “There are two two-center two-electron σ bonds”

Is Fig. 3 an ORTEP or is it really a Mercury plot with a partial thermal ellipsoids? Either way, the crystallographic section of the SI or the main text should have the appropriate reference for the graphic generation.

Response: This figure was rendered using the program OLEX2. The statement “ORTEP depiction” has been changed to “Thermal ellipsoid plot” Additionally the statement “Structures were rendered using OLEX 2 version 1.3 at 30% probability ellipsoids.” was added to the supplemental information with reference to doi: 10.1107/S0021889808042726

Fig. 4 should have the scan rate listed.

Response: The statement “ $\nu = 100 \text{ mV s}^{-1}$ ” has been added to Figure 4.

Values in Figure 4 caption and two separate places in the text (line 100 and 109) text do not match. -0.94 vs -0.95 V. Please clarify the correct value or if these are in reference to different scan rates

Response: The assignment of $E_{1/2} = -0.95$ has been corrected to $E_{1/2} = -0.94$.

We thank the reviewer for their thorough and detailed evaluation of our work and we appreciate their support.

Reviewer #4 (Remarks to the Author):

The authors report a very rare example of a complex presenting a Ce(IV)-C bond and the first Ce(IV)-aryl complex by using an original synthetic approach.

This finding is of high interest to the community and should be published in Nature Communications.

However, it is not evident to me why the authors define the ^{13}C carbon shift at 255 ppm anomalous since it falls in the range of reported Ce(IV)-C shifts ranging from 213 to 343 ppm. Please clarify this aspect in the manuscript.

Response: The anomalous shift is in reference to the shifts of other diamagnetic M(IV)-Ar species with reference to the Th(IV) shifts reported in the manuscript. To further clarify this message, we have added this statement: “In this light **3-THF** and **3-DME** has an anomalously high covalency for a M^{IV} -Aryl interaction.”

Moreover, the computational studies fail overall to fully distinguish if the carbon shift is due to covalency or to the presence of multiconfigurational states.

Response: To address this comment, in the revised manuscript, we have presented natural localized molecular orbital analyses of the ^{13}C shielding (see Table 2 and the last two paragraphs before the Discussion Section). The analysis shows clearly that the Ce- C_{aryl} covalency does play an important role on the observed chemical shift. A ground-state with slight multiconfigurational character could be present and we believe that is the reason for the

sensitivity of the calculated ^{13}C shift to the applied DFT approximation. CASSCF calculations and additional spectroscopic studies are under way to confirm or refute this assumption and will be presented in a separate work.

I would suggest to perform additional spectroscopic studies (VT magnetic studies or Xanes) to further assess the presence of multiconfigurational studies that would make harder to pinpoint the effects of covalency on the ipso carbon chemical shifts. It would be helpful to further elucidate how the carbon shift can be related to covalency if it results from a mix of electronic configurations.

Response: Our intention is that these experiments will be included in a future publication. In line with Reviewer 3's comment "I would suggest that the synthetic and theoretical efforts described here be published as one complete story and the further necessary spectroscopic, magnetometry, and theoretical studies be published separately." For the current work, we have instead provided a more in depth look into the origin of the shift computationally and have found that the shift can be explained by the sizable Ce-C_{aryl} covalency and SOC effects rather than invoking mixed electronic configurations. See **Table 2**.

Minor points

Page 6 line 107: please provide a reference when mentioning the E_{1/2} value of the Ce(III)/Ce(IV) couple for **1**

Reference to doi: 10.1021/ic400202r has been added to the statement "which is the Ce^{III}/Ce^{IV} couple of **1**."

Page 7 line 109: give reference when mentioning reported CV of **2**

The statement "consistent with previous assignments for compounds **1** and **2** (Figure 4)." Has been changed to "consistent with the previous assignment for compound **1** and inferred for compound **2**" because **2** is not stable to the electrochemical conditions.

We thank the reviewer for their careful consideration of the electrochemical data.

REVIEWERS' COMMENTS

Reviewer #1 (Remarks to the Author):

This manuscript has been revised and the result is a nice contribution to Ce(IV) chemistry! The authors have addressed the concerns of the reviewers well, and I support its publication.

Reviewer #2 (Remarks to the Author):

My concerns have been adequately addressed.

Reviewer #3 (Remarks to the Author):

The authors have completed a thorough and attentive revision. The manuscript is ready for publication in its current form.

Reviewer #4 (Remarks to the Author):

The authors have fully addressed my comments and those of the other reviewers in a rigorous and careful matter.

In my opinion this very interesting manuscript is now ready for publication.

Reviewer #1 (Remarks to the Author):

This manuscript has been revised and the result is a nice contribution to Ce(IV) chemistry! The authors have addressed the concerns of the reviewers well, and I support its publication.

Reviewer #2 (Remarks to the Author):

My concerns have been adequately addressed.

Reviewer #3 (Remarks to the Author):

The authors have completed a thorough and attentive revision. The manuscript is ready for publication in its current form.

Reviewer #4 (Remarks to the Author):

The authors have fully addressed my comments and those of the other reviewers in a rigorous and careful matter.
In my opinion this very interesting manuscript is now ready for publication.

We again thank the reviewers for their diligent work in improving the quality of this manuscript.